# Ophthalmic Complications after Needle-Based Medical Aesthetic Procedures: A Narrative Review

**DOI:** 10.3390/jcm12010313

**Published:** 2022-12-31

**Authors:** Karolina Bonińska

**Affiliations:** Miejskie Centrum Medyczne Jonscher, ul. Milionowa 14, 93-113 Łódź, Poland; okulistyka@jonscher.pl or karolina.boninska@gmail.com; Tel.: +48-607816193

**Keywords:** aesthetic medicine, botulinum toxin, foam sclerotherapy, hyaluronic acid, iatrogenic complications

## Abstract

Background: This study aimed to discuss common complications of medical aesthetic treatments, which require ophthalmological intervention. Methods: This literature study evaluated published journal articles (clinical trials or scientific reviews) that were extracted from electronic databases (MEDLINE and PubMed) and reference lists of related articles. Only articles available in English were considered for this review. Results: Unskillful interference in the eye area can cause severe, irreversible complications, including blindness. This is a constant risk because of anatomical deviations, and retrograde blood flow. Conclusions: Accurate knowledge of anatomy, especially the vascular anatomy of high-risk sites, and the understanding of the depth and plane of injection, and various injection techniques minimize the risk of these complications.

## 1. Introduction

Anti-aging therapies for the orbital area are difficult due to the unique properties of the skin in this area, and anatomical conditions that increase the risk of complications. Rich vascularity around the eye is inimical to needle treatments, and the recovery period after any intervention is much longer than in other parts of the face. Moreover, unskillful interference with the eye area can cause severe, irreversible complications, including blindness.

This study aimed to discuss the most common complications of medical aesthetic treatments, which require the intervention of an ophthalmologist. Presumably, there have been no such reports.

### Literature Search

This literature research assessed published journal articles (clinical trials or scientific reviews) up to November 2022. Studies were identified by searching electronic databases (MEDLINE and PubMed) and reference lists of related articles. Only articles available in English were considered for this review. The following keywords were used in various combinations to extract relevant studies: ocular complications after aesthetic medicine, aesthetic medicine complications, botulinum toxin complications, foam sclerotherapy complications, hyaluronic acid complications, and facial injection. In addition, the references cited in the identified articles were reviewed to find any additional reports.

## 2. Complications after Procedures with the Use of Botulinum Toxin

Notably, it was in ophthalmology that the botulinum toxin type A (BTX-A) [1,2] preparation—most commonly utilized in aesthetic medicine—was originally used. It works by blocking the release of acetylcholine at the level of the neuromuscular junction of skeletal muscles, resulting in reversible paralysis. Although randomized, multicenter studies confirm its safety [3], the following complications may arise when it is injected improperly.

### 2.1. Drooping of the Upper Eyelid (Ptosis)

Ptosis, resulting from the use of botulinum toxin, occurs in 0.71% of cases [4] 3–14 days after the injection [5]. It appears when the preparation reaches the levator muscle of the upper eyelid.

To prevent the occurrence of ptosis, administering too much botulinum toxin and diluting its preparation excessively should be avoided. Another factor resulting in ptosis is injecting too close to the upper edge of the orbit. All these factors favor the diffusion of the neurotoxin into the tissues [4]. 

Drooping of the eyelid ceases spontaneously after 3–6 months. To shorten this time, 0.5% apraclonidine drops, an alpha 2-adrenergic agonist, can be used to lift the eyelid by 1–3 mm by stimulating the contraction of the Muller’s muscle. The most common dosage is 3 drops daily until the eyelid ceases drooping. Other possible medications are 0.1% or 0.2% brimonidine, 2.5% neosynephrine hydrochloride, and 0.1% naphazoline drops. These substances also increase the tone of Muller’s muscle [6,7]. 

### 2.2. Strabismus

Strabismus after injection of botulinum toxin is a rare complication, occurring in approximately 1.7% of cases [8]. It arises as a result of paresis or paralysis of an extraocular muscle, which manifests as eye misalignment, especially when looking in the direction of action of the affected muscle. This complication occurs when the toxin is administered in excessive amounts and/or injected too close to the upper edge of the orbit. It then diffuses into the extraocular muscles—usually the inferior oblique, which lies at the lower edge of the orbit [9]. This complication occurs with deep injections into the central part of the lower eyelid [9].

Strabismus can be corrected with prismatic glasses until the toxin is broken down in the tissues.

### 2.3. Damage to the Eye Surface

Botulinum toxin may cause eye surface disorders. Ozgur et al. described complications that occur in the case of repeated injections into the lateral canthal region for correction of crow’s feet [10]. They indicated the following initial mild symptoms (1+): eye irritation with foreign body sensation. If the injection of botulinum toxin is continued in this region, the symptoms intensify (2+). Subsequently, swelling of the eyelids, increased tearing, and exposure of the sclera can be observed. Serious (3+) symptoms include: considerable tearing, ectropion, photophobia, pain, corneal erosion, and fatty hernia of the lower eyelid [10].

The mechanism that is responsible for the creation of the aforementioned disorders is chemo-denervation of the orbicularis oculi, resulting in impaired blinking, lagophthalmos, and ectropion, further resulting in eye irritation with foreign body sensation and excessive tearing. The authors also indicate the possibility of multifactorial pathological tearing, including dry eye syndrome, causing reflex tearing; hypotonicity of the medial pretarsal fibers causing a decreased outflow of tears, and malposition of the eyelids causing impaired retention of tears, as well as BTX-A diffusion into the lacrimal gland diminishing tear production and further aggravating the dry eye [10].

Mild (1+) symptoms are treated by ophthalmologists with eye lubricants or artificial tears. If symptoms worsen (2+), treatment needs to be intensified. If symptoms persist (3+), lateral musculoplasty of the lower eyelid [10] is often required.

### 2.4. Eyeball Perforation

Careless administration of the preparation may wound or penetrate the eyeball. After an inadvertent administration of botulinum toxin into the eyeball described in the literature [11], the intraocular pressure increased to 50 mmHg with mydriasis and tearing of the retina, which led to its detachment. In the above case, 7.5 units of type A botulinum toxin with a volume of 0.15 mL, were administered. Intraocular pressure-lowering drugs were administered locally (pilocarpine and timolol drops) along with 500 mg of intravenous acetazolamide, and laser therapy was performed to prevent retinal tearing. Full visual acuity was restored after 2 weeks [11]. 

This case confirms the lack of toxicity of botulinum toxin to the retina. The increase in intraocular pressure resulted from the increase in the volume of the eyeball, although the mechanism of acute closure of the iridocorneal angle was also possible [11].

## 3. Complication after Procedures with the Use of Hyaluronic Acid

Hyaluronic acid as a soft tissue filler has become increasingly popular over the last two decades due to its immediate effect and the reversibility of the procedure—thanks to hyaluronidase. The most serious complication after its application is the occlusion of the arterial vessel [12,13,14]. Aspiration before filler injection does not protect against this complication [15]. 

### Central Retinal Artery Occlusion

Hyaluronic acid can lead to the closure of the central retinal artery, which is anatomically and functionally the terminal artery. This very serious complication occurs due to vascular branches of the internal carotid artery supplying both the face and the eye, and through retrograde blood flow of the ophthalmic arterial branch [14,16,17]. Particular attention must be paid to the glabella (supratrochlear and supraorbital arteries) region, nasal (lateral and dorsal nasal arteries) region, nasolabial fold (angular artery), forehead (supratrochlear and supraorbital arteries) and temple area (superficial temporal artery) [18,19]. Moderate risk sites include the periocular region and cheek [19]. Coleman measured the threshold volume of emboli required to fill the ophthalmic artery as 0.1 mL [13,20]. Khan et al. measured the volume of the supratrochlear artery as only 0.085 mL [13,21].

The first symptom reported by the patients is a significant decrease in visual acuity, including no light perception. Other symptoms reported were ocular pain, headache, nausea, and vomiting [19]. Its recovery is determined by the rapidity of the treatment provided by the ophthalmologist. It includes lowering the intraocular pressure, massaging the eyeball, or puncturing the anterior chamber to push through the embolic material. The prognosis for vision return is more questionable, the longer the obstruction lasts.

In addition to central retinal artery occlusion symptoms, lack of extraocular movement, ophthalmoplegia, persistent ptosis, exotropia, and skin necrosis were also noted [19].

## 4. Complications after Foam Sclerotherapy 

The two sclerosants most commonly used in foam sclerotherapy are sodium tetradecyl sulfate (STS) and polidocanol. Their task is to damage the vascular endothelium and induce the process of fibrosis with subsequent vasoconstriction. Although sclerotherapy is a procedure with a broad safety profile [22], the literature describes cases of complications bordering on ophthalmological and neurological diseases.

### 4.1. Visual Field Defects

Hartmann et al. described complications after the administration of 9 mL of 3% polidocanol under ultrasound control in the treatment of varicose veins of the lower extremities (left great saphenous vein and a truncal varicosity of the left small saphenous vein). Immediately after the injection, the patient reported photopsia which ceased after a few minutes. Two hours later, symptoms of dysphasia were noted [23]. Neurological examination, cranial magnetic resonance, electrocardiography, and thrombophilia findings were unremarkable. Multiplanar transoesophageal echocardiography revealed a persistent foramen ovale [23].

A possible pathomechanism of complications includes an air bubble embolism from the sclerosant. The coexistence of the foramen ovale is conducive to the displacement of the embolic material into the cerebral circulation. 

The visual disturbances after foam sclerotherapy could be a warning sign of potential neurological deficits. If these persist, treatment with hyperbaric oxygen is the preferred form of therapy [24].

### 4.2. Central Retinal Artery Occlusion

The literature also describes a case of an embolism of the central retinal artery that occurred after the injection of 1 mL of STS under ultrasound guidance due to prominent facial veins (supratrochlear vein) on the forehead [25]. The patient in question reported a sudden, painless, unilateral decrease in visual acuity after the procedure. The ophthalmoscopic examination revealed the characteristic symptoms of central retinal artery occlusion, confirmed by imaging tests. Moreover, afferent pupillary defects and mydriasis occurred. In the above case, 500 mg acetazolamide intravenously, carbon dioxide (paper bag) rebreathing, ocular massage, and anterior chamber paracentesis were used. Despite the prompt intervention, visual acuity remained at no light perception with the persistent afferent pupillary defect. During the observation period, ischemic retinopathy occurred, requiring laser panretinal photocoagulation.

The likely cause of the described complication was an unintentional injection into the supratrochlear artery, whereby the sclerosant entered the ophthalmic artery and its end branches including the central retinal artery by the retrograde flow. This led to vascular occlusion of the retinal and choroidal circulations with consequent ischemia occurring within 30 min. Imaging and angiography confirmed that the patient had no anatomical variants predisposing her to this complication [25]. 

### 4.3. Orbital Infarction Syndrome

This rare ischemic syndrome manifested as ptosis, exophthalmos, and blindness after the 14th injection, in an 11-year-old child with facial vascular malformation [26]. In this case, 2 mL of 3% STS was administered under ultrasound guidance. Shortly after waking up, the patient reported severe eye pain with an inability to open the eyelid. The doctor indicated ophthalmoplegia, with one-sided dilation of the pupil. Computed tomography angiography revealed a lack of contrast flow in the ophthalmic artery, the superior ophthalmic vein, and the extraocular muscles. After 4 months, the eyelid ceased drooping, and the mobility of the eyeball returned to normal. Unfortunately, the patient remained blind; the eyeball also atrophied secondarily [26].

The authors suspect that the resulting ischemia was due to the occlusion of the central retinal artery affected by the stenosis or occlusion of the internal carotid artery and its branches [26,27]. Treatment of the syndrome includes anticoagulants, systemic steroid therapy, and hyperbaric therapy [27]. Unfortunately, the prognosis is always unfavorable. 

## 5. Discussion

Owing to the growing demand for medical aesthetic procedures, ophthalmologists should be familiar with the procedures performed in the orbital area to react properly in the event of complications. Furthermore, it is extremely important to obtain a thorough history because patients may not remember or admit to undergoing previous anti-aging treatments. 

Appropriate qualifications are required to perform risky procedures. To avoid possible complications, the ability to perform appropriate injection techniques and a thorough knowledge of the effects of the used preparations are mandatory [19]. Only aesthetic doctors, who have undergone prior training, have the appropriate knowledge and skills to correct physical defects by performing invasive procedures. The doctor performing the procedure is also responsible for the treatment of possible side effects resulting from his/her intervention. The patient should know what product the doctor intends to use, and what is its safety profile or origin. He/she should also be informed about all the consequences of using a given form of therapy, including the degree, and the possible range of complications.

Understanding facial anatomy and its variations are crucial to minimizing the risk of vascular complications [28,29,30]. Beleznay et al. have discussed in detail high-risk anatomic sites of injection with limited collateral perfusion [19]. The ophthalmic artery, a branch of the internal carotid, divides into 10 vessels including the central retinal, supraorbital, supratrochlear, dorsal nasal, and lacrimal arteries. These arteries could be compromised when injecting the glabella, nose, and forehead. If the cannula perforates an artery and excessive pressure is applied when injecting the filler, the arterial resistance may decrease, and the filler could embolize the ocular vessels [19].

An interesting concern relating to hyaluronic acid is aspiration after injection. Goodman et al. question the value of this procedure as an assurance of safety [15]. When a practitioner is aspirating for a prolonged time, he applies negative pressure on the syringe which may shift the needle position and move the tip. This makes the aspiration unreliable. Patient movement, even minute reactive or mimetic actions, can also shift the placement of the instrument tip. Staying still once in position does not guarantee a good outcome. Injectors who aspirate rely on the test’s predictive power to avoid injecting in vessels. The potential for false negative aspirations means the test can give an injector a false sense of safety and lead to adverse vascular events. Better safety solutions to prevent vascular compromise include: continuous needle movement, slow injection, low plunger pressure, small doses, and a sufficient understanding of safe injection sites and vascular anatomy [15].

Furthermore, hyaluronidase is effective in degrading hyaluronic acid and reversing vascular occlusion [31,32]. However, blindness from hyaluronic acid-filler embolisms in the central retinal artery remains an unsolved problem because of the difficulty of administering the medication into the retinal artery [31,32].

## 6. Conclusions

This article describes three types of needle procedures in the field of aesthetic medicine. In the case of botulinum toxin injection, the complications that arise are usually transient until the toxin breaks down in the tissues. Appropriate patient selection and the injection site are crucial. After the injection, the patient should be informed about these post-treatment complications.

Injections of hyaluronic acid in the facial area carry the risk of vascular occlusion, leading to skin necrosis or blindness. Accurate knowledge of anatomy, and various injection techniques, as well as the differences in the available preparations, minimize the risk of these complications. Anatomical deviations, and retrograde blood flow mean that the risk is constant. 

Foam sclerotherapy complications can be reduced by controlling points of reflux with compression of the treated area. To avoid arterial embolization, ultrasound-guided injections, leveling the patient for 5 min after injection using small boluses to limit retrograde flow, and maintaining low pressure injections with the lowest possible concentrations of sclerosant or drug [25] are recommended.

Further studies should be conducted to investigate complications after other types of medical aesthetic procedures.

## Data Availability

Not applicable.

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
