# Peer review of "Ophthalmic Complications after Needle-Based Medical Aesthetic Procedures: A Narrative Review"

_jcm, 2022, doi:10.3390/jcm12010313_

Round 1
Reviewer 1 Report
Authors did not adress the role of using hyaluronidase in treating arterial occlusion affecting the eye from fillers. It should be adressed, and if in-effective it would be important to clarify this.
Author Response
Reviewer 1
- Authors did not adress the role of using hyaluronidase in treating arterial occlusion affecting the eye from fillers. It should be adressed, and if in-effective it would be important to clarify this.
Reply: Thank you for your suggestion. The comment has been added to the Discussion.
“Furthermore, hyaluronidase is effective in degrading hyaluronic acid and reversing vascular occlusion [31,32]. However, blindness from hyaluronic-acid-filler embolisms in the central retinal artery remain an unsolved problem because of the difficulty of administering the medication into the retinal artery [31,32].”

Reviewer 2 Report
The author aims to review the ophthalmic complications related to needle-based aesthetic procedures in the face area, focusing especially on BTX-A injection, dermal fillers and foam sclerotherapy.
While the concept is interesting and the examples reported are well-described and scientifically sound, the review lacks of originality and scientific relevance compared to other works already available in the literature (for example: https://doi.org/10.1016/j.survophthal.2021.04.009).
The proposed review is generic and superficial, and describes only a few anecdotical cases. The research methodology is described only in the abstract, and less than 20 papers are cited for a review of a topic of great interest in the aesthetic medicine community. 10 of these papers were published before 2010.
As an example, in paragraph 3 no reference to the use of hyaluronidases to dissolve HA fillers has been made and only one case-report is presented. In the “discussion” and “conclusion” paragraphs, several references are made by the author on the importance of selecting “highest quality products” and on the knowledge of anatomy. However the author never describes in the review how the quality of the product influence the probability of serious side-effects nor the author describes which anatomical deviations can be relevant and if or how the doctor can check for anatomical deviations.
In line 110, the author writes “Aspiration before injection does not protect against this complication”. However, no references are made to substantiate this sentence.
In the conclusions, line 207, the author writes: “Accurate knowledge of the anatomy, and various injection techniques, as well as the differences in the available preparations minimize the risk of these complications.”
However, no reference on the possible injection techniques or on the available products in the market are made.
In general, the review appears superficial and outdated. An extensive literature research and a more detailed and in-deep discussion is needed before the paper will be ready to be published.
Author Response
Reviewer 2
- The author aims to review the ophthalmic complications related to needle-based aesthetic procedures in the face area, focusing especially on BTX-A injection, dermal fillers and foam sclerotherapy. While the concept is interesting and the examples reported are well-described and scientifically sound, the review lacks of originality and scientific relevance compared to other works already available in the literature (for example: https://doi.org/10.1016/j.survophthal.2021.04.009).
The proposed review is generic and superficial, and describes only a few anecdotical cases. The research methodology is described only in the abstract, and less than 20 papers are cited for a review of a topic of great interest in the aesthetic medicine community. 10 of these papers were published before 2010
Reply: Thank you for your comment. My study describes only eye-threatening complications after needle-based procedures from ophthalmologist point of view. I have changed my manuscript according to your suggestions.
- As an example, in paragraph 3 no reference to the use of hyaluronidases to dissolve HA fillers has been made and only one case-report is presented.
Reply: Thank you for your comment. These parts have been changed as suggested.
“Furthermore, hyaluronidase is effective in degrading hyaluronic acid and reversing vascular occlusion [31,32]. However, blindness from hyaluronic-acid-filler embolisms in the central retinal artery remain an unsolved problem because of the difficulty of administering the medication into the retinal artery [31,32].”
- 3. In the “discussion” and “conclusion” paragraphs, several references are made by the author on the importance of selecting “highest quality products” and on the knowledge of anatomy. However the author never describes in the review how the quality of the product influence the probability of serious side-effects nor the author describes which anatomical deviations can be relevant and if or how the doctor can check for anatomical deviations.
Reply: Thank you for your comment. I have changed “discussion” and “conclusion” part. In regard to anatomic variation, I have decided to cited the latest literature which described individual variability with the atlas accuracy. In my option there is no safe area. Some of the facial region need special attention.
“Understanding facial anatomy and its variations is crucial to minimize the risk of vascular complications [28-30]. Beleznay et al have discussed in detail high risk anatomic sites of injection with limited collateral perfusion [19]. The ophthalmic artery, a branch of the internal carotid, divides into 10 vessels including the central retinal, supraorbital, supratrochlear, dorsal nasal, and lacrimal arteries. These arteries could be compromised when injecting the glabella, nose, and forehead. If the cannula perforates an artery and excessive pressure is applied when injecting the filler, the arterial resistance may decrease and the filler could embolize the ocular vessels [19].”
- In line 110, the author writes “Aspiration before injection does not protect against this complication”. However, no references are made to substantiate this sentence.
Reply: Thank you for your suggestion. I have revised the text and cited the appropriate reference.
“Aspiration before fillers injection does not protect against this complication [15].”
“An interesting concern relating to hyaluronic acid is aspiration after injection. Goodman et al question the value of this procedure as an assurance of safety [15]. If there is any doubt about negative aspiration, there is no point in using it. Achieving a positive aspiration would postpone the risk until the next injection and negative aspiration would still be needed [15]. Better safety solutions to prevent vascular compromise include: continuous needle movement, slow injection, low plunger pressure, small doses, and a sufficient understanding of safe injection sites and vascular anatomy [15].”
- In the conclusions, line 207, the author writes: “Accurate knowledge of the anatomy, and various injection techniques, as well as the differences in the available preparations minimize the risk of these complications.” However, no reference on the possible injection techniques or on the available products in the market are made.
Reply: Thank you for your comment. I have added proper comment.
“An interesting concern relating to hyaluronic acid is aspiration after injection. Goodman et al question the value of this procedure as an assurance of safety [15]. If there is any doubt about negative aspiration, there is no point in using it. Achieving a positive aspiration would postpone the risk until the next injection and negative aspiration would still be needed [15]. Better safety solutions to prevent vascular compromise include: continuous needle movement, slow injection, low plunger pressure, small doses, and a sufficient understanding of safe injection sites and vascular anatomy [15].”
- In general, the review appears superficial and outdated. An extensive literature research and a more detailed and in-deep discussion is needed before the paper will be ready to be published.
Reply: I have revised the manuscript according to the suggestions provided. I hope you find my revisions satisfactory. I look forward to hearing from you and would be happy to make further changes, if required.

Round 2
Reviewer 1 Report
Paper is improved, well done!
Author Response
Thank you for your comment.
Reviewer 2 Report
I appreciate the response from the author and I acknowledge a significant improvement in the English, the readability and the scientific value. However, some issues remains.
Regarding the negative aspiration, reference 15 is cited three times in one paragraph and the full paragraph is just a rephrasing of the conclusion of reference 15. The rephrasing also seems to slightly alter sense of the reference 15 "Achieving a positive aspiration would just defer the risk to the next injection location where a negative aspiration would then be relied on." is different from "Achieving a positive aspiration would postpone the risk until the next injection and negative aspiration would still be needed".
I suggest that the author improve this paragraph by discussing reference 15 in more details and more clearly, adding also a wider context and better explaining the scientific discussion on this point.
Author Response
Reviewer 2
I appreciate the response from the author and I acknowledge a significant improvement in the English, the readability and the scientific value. However, some issues remains.
Regarding the negative aspiration, reference 15 is cited three times in one paragraph and the full paragraph is just a rephrasing of the conclusion of reference 15. The rephrasing also seems to slightly alter sense of the reference 15 "Achieving a positive aspiration would just defer the risk to the next injection location where a negative aspiration would then be relied on." is different from "Achieving a positive aspiration would postpone the risk until the next injection and negative aspiration would still be needed".
I suggest that the author improve this paragraph by discussing reference 15 in more details and more clearly, adding also a wider context and better explaining the scientific discussion on this point.
Reply: Thank you for your comment. This part has been changed as suggested:
“An interesting concern relating to hyaluronic acid is aspiration after injection. Goodman et al. question the value of this procedure as an assurance of safety [15]. When a practitioner is aspirating for a prolonged time, he applies a negative pressure on the syringe what may shift the needle position and move the tip. It makes the aspiration unreliable. Patient movement, even minute reactive or mimetic actions, can also shift the placement of the instrument tip. Staying still once in position, does not guarantee a good outcome. Injectors who aspirate rely on the test’s predictive power to avoid injecting in vessels. The potential for false negative aspirations means the test can give an injector a false sense of safety and lead to adverse vascular events. Better safety solutions to prevent vascular compromise include: continuous needle movement, slow injection, low plunger pressure, small doses, and a sufficient understanding of safe injection sites and vascular anatomy [15].
